# HAIR: Head-mounted AR Intention Recognition

### David Puljiz
Intelligent Process Automation and Robotics Lab (IPR),
Institute for Anthropomatics and Robotics, Karlsruhe
Institute of Technology
Karlsruhe, Germany
david.puljiz@kit.edu

### Bowen Zhou
Intelligent Process Automation and Robotics Lab (IPR),
Institute for Anthropomatics and Robotics, Karlsruhe
Institute of Technology
Karlsruhe, Germany

### Ke Ma
Intelligent Process Automation and Robotics Lab (IPR),
Institute for Anthropomatics and Robotics, Karlsruhe
Institute of Technology
Karlsruhe, Germany

### Björn Hein
Intelligent Process Automation and Robotics Lab (IPR),
Institute for Anthropomatics and Robotics, Karlsruhe
Institute of Technology
Karlsruhe, Germany

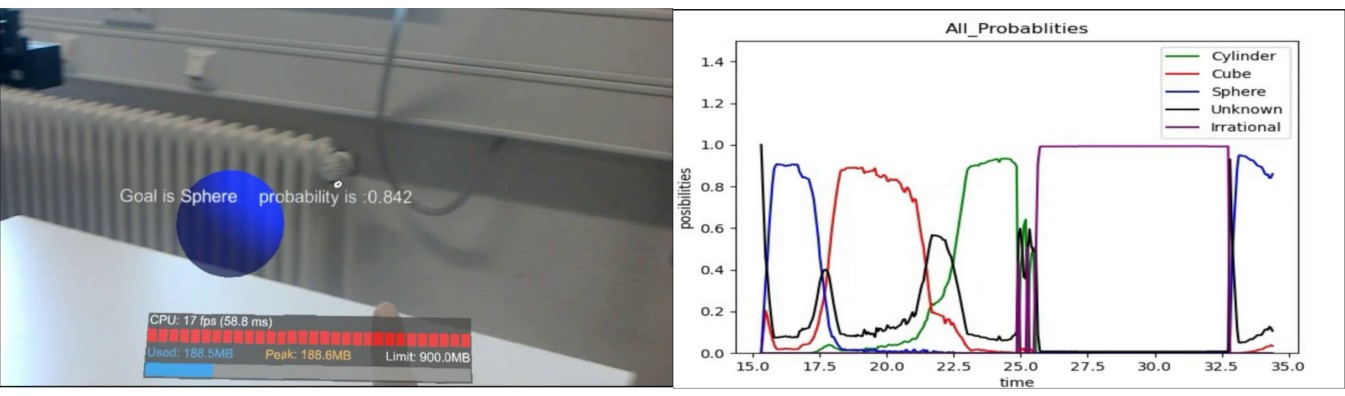

**Figure 1: The view from the HoloLens on the left, showing the user looking at the goal "Sphere" and approaching it with the hand. On the right the current and previous probabilities. The user rotated from the first goal, the sphere, through the other two goals and back towards the sphere. To note is how the output proclaims the user irrational when they are facing away from all the defined goals**

## ABSTRACT

Human teams exhibit both implicit and explicit intention sharing. To further development of human-robot collaboration, intention recognition is crucial on both sides. Present approaches rely on a vast sensor suite on and around the robot to achieve intention recognition. This relegates intuitive human-robot collaboration purely to such bulky systems, which are inadequate for large-scale, real-world scenarios due to their complexity and cost. In this paper we propose an intention recognition system that is based purely on a portable head-mounted display. In addition robot intention visualisation is also supported. We present experiments to show the quality of our human goal estimation component and some

basic interactions with an industrial robot. HAIR should raise the quality of interaction between robots and humans, instead of such interactions raising the hair on the necks of the human coworkers.

## KEYWORDS

Human Intention Estimation, Augmented Reality, Human-robot Collaboration, Head Mounted Displays

**ACM Reference Format:**
David Puljiz, Bowen Zhou, Ke Ma, and Björn Hein. 2021. HAIR: Head-mounted AR Intention Recognition. In ,. ACM, New York, NY, USA, 6 pages. https://doi.org/10.1145/nnnnnnn.nnnnnnn

## 1 INTRODUCTION

Communicating intentions between members of a team is paramount for successful cooperation and task completion. Previous work in the field of Augmented reality (AR) human-robot interaction (HRI) focused on either improving robot programming [13] or visualising robot motions [14]. Although quite important for collaboration, such systems still lack the estimation of the human intention from the robot's side. Several such systems have been proposed, such as [1] where the human is tracked and their goal

*VAM-HRI '21, Virtual, Augmented and Mixed reality human-robot interaction Workshop*
© 2021 Association for Computing Machinery.
ACM ISBN 978-x-xxxx-xxxx-x/YY/MM...$15.00
https://doi.org/10.1145/nnnnnnn.nnnnnnn

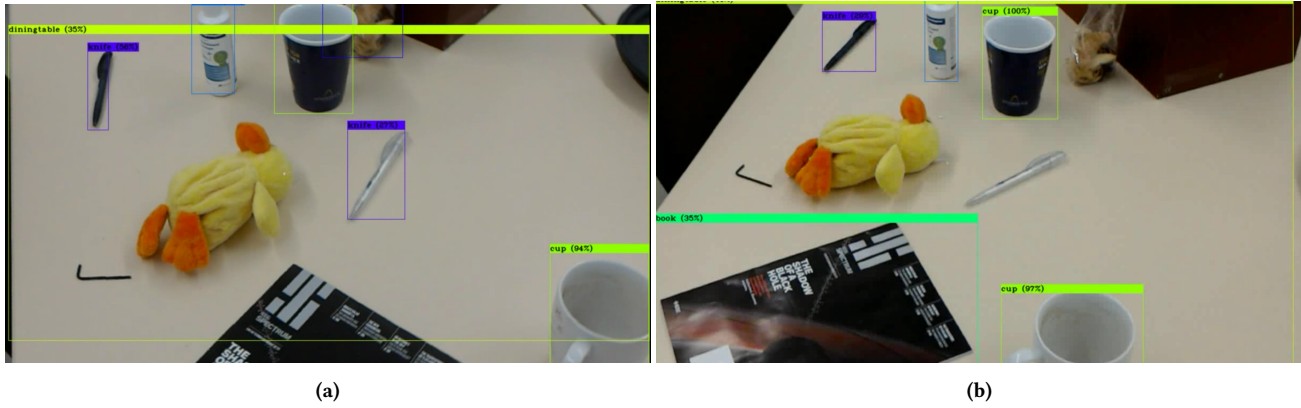

(a)  (b)

**Figure 2: YOLOv4 classification of HoloLens camera data. Bounding box classification approaches together with the known HoloLens egomotion and depth data can be used to define the interaction objects/goals as well as to outsource part of the environmental sensing from the robot to the human.**

estimated inside a robot cell. Such systems however require a big overhead in complexity and cost of the robot cells. With the advent of Head-mounted Displays (HMDs) the possibility arises of a fully portable, completely worn system possessing both the robot intention visualisation and human intention estimation.

A similar system based on a HMD and intended for human-robot collaborative task planning was presented by Chakraborti et al. [3]. In the presented system, however, the human coworker had to explicitly select and reserve objects it wished to interact with, slowing down task execution and increasing physical and mental workloads on the human worker.

Here we propose a system that implicitly evaluates the intentions of the human, thus minimising the increase in workload. The proposed system is based on the Microsoft's HoloLens HMD and is aimed at a collaborative scenario between a single human and an industrial manipulator. The system is robot agnostic and completely portable requiring a very short set-up at the beginning of interaction. This guarantees that a single human worker can interface with multiple robots one after another, without the need for specialised robot cells or sensors around any of those robots.

The system takes as input the pose of the HMD in the world coordinate system, the position of the hand joints on the world coordinate system as well as a set of possible spatial goals, which can be added and removed during the interaction itself. The output is a set of probabilities of the goal the human wants to approach as well as the action they wish to perform.

This paper will present our current work and tests aimed mostly at having a robust goal estimation. To the best of the authors knowledge such an intention estimation algorithm using a completely worn system has not yet been developed.

## 2 METHODOLOGY

### 2.1 Referencing

First and foremost a common coordinate system must be establish between the HMD and the robotic manipulator. Referencing can be done in a variety of ways, perhaps the most popular is the use of

QR codes or other preset visual markers [6]. Although these offer continuous instead of one-shot referencing, as well as very good precision, they require a setup step that we would like to avoid. Manually selecting the robot base such as presented in [13] is more flexible yet also more imprecise. We have proposed several referencing methods in [12], with the semi-automatic one, consisting of a rough user guess followed by a refinement step, offering the best balance between accuracy and computational time. The refinement step consists of filtering a point cloud captured with the HMD and using a registration algorithm to fit the model of the manipulator into the filtered point cloud, using the user guess as the start point of the registration algorithm. The user guess prevents the common problem of registration algorithms being stuck in the local minima, and we found that even a basic ICP algorithm performs a good job of refining the guess of the user. Another approach is a fully automatic one without a user guess. Such a referencing algorithm, similarly to our automatic method proposed in [12], was proposed by Ostantin et al. [8]. It clusters the point cloud captured by the HMD using the DBSCAN clustering algorithm and then performs model matching between the clusters and a model of the robot.

### 2.2 Defining Spatial Goals

Secondly the possible spatial goals of the human and the robot need to be defined. In case the robot does not possess the full map of its surroundings, the HMD can also provide that as we demonstrated in [11]. This can also include possible goals and regions of interest such as the table or the conveyor belt. If goals are specific objects, here too the HMD can provide types and positions of those objects. One such possibility is through the use of bounding-box classifiers such as YOLOv4 [2]. In Fig. 2, one can see the result of running YOLOv4 on the HoloLens camera data. Having the egomotion data of the hololens, as well as data from the HMD's depth sensor, allows a full spatial definition of the objects and therefore possible goals.

The user should also be able to add and remove goals manually during the interaction step. Therefore the goal estimation algorithm was selected to allow such a modality.

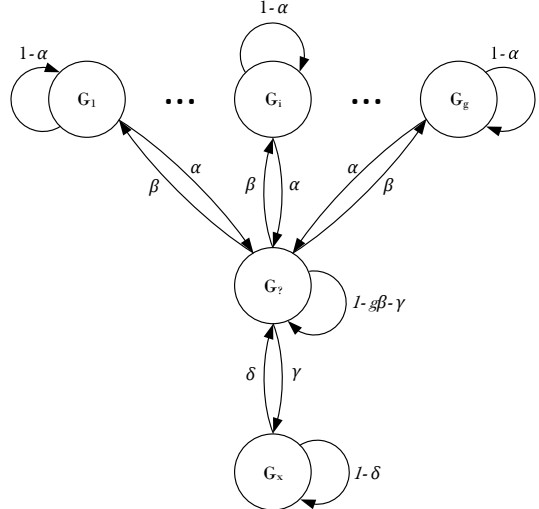

**Figure 3: The HMM states of the human goal intention estimation system as presented in [10]. It consists of g goal states, a state of unknown intention $G_?$ and a state of the human acting irrationally $G_x$.**

## 2.3 Goal Estimation

Finally, having a common coordinate system, mapped working environment and possible goals, one can infer the goals using a human intention recognition algorithm. We base our HIR algorithm on previous work by Petkovic et al. [10] where a hidden Markov model framework was used to estimate the goal of the human in an automated warehouse. The approach is quite general and with minimal modification can be adapted to be used in our use case. In this section we will present a brief overview of the calculation, for more details please refer to the original paper.

Instead of the position of the human coworker as in the original paper, we consider the position of the hand in relation to goal objects. To simplify the calculations, we assume there is an almost straight line between the hand position and each goal object. By doing that we can forego the complex path planning step to determine the modulated distance and instead use the euclidean distance to calculate the vector $\mathbf{d}$ that represents the distance of the hand to each goal. As in [10], we define additional 32 points $p_i$ on a circle around the previous hand position $l'$ and a radius $r$ equal to the distance between the current $l$ and the previous $l'$ hand positions. We calculate the vector $\mathbf{d}$ for each point $p_i$ and append them to the modulated distance matrix $\mathbf{D}$.

Additionally we consider the gaze validation $\mathbf{s}$ of the HMD. The motivation being that the user is more likely to look approximately towards the goal of the hand motion than towards other goals. The gaze validation is calculated as:

$$\mathbf{s}_i = \begin{cases} \mathbf{g} \cdot \dfrac{\mathbf{o}_i - \mathbf{h}}{||\mathbf{o}_i - \mathbf{h}||}, & \mathbf{g} \cdot \dfrac{\mathbf{o}_i - \mathbf{h}}{||\mathbf{o}_i - \mathbf{h}||} \geq 0 \\ 0, & \text{otherwise.} \end{cases} \quad (1)$$

Where $\mathbf{g}$ is the HMD orientation in the world coordinate system, $\mathbf{o}_i$ is the position of object i and $\mathbf{h}$ is the position of the HMD. We expand the motion validation vector $\mathbf{v}$ as follows:

$$\mathbf{v} = \frac{\max\limits_{1 \leq i \leq n} \mathbf{D}_{ij} - \mathbf{d}}{\max\limits_{1 \leq i \leq n} \mathbf{D}_{ij} - \min\limits_{1 \leq i \leq n} \mathbf{D}_{ij}} \cdot \mathbf{s} \quad (2)$$

The rest follows exactly the algorithm described in [10]. We use the same transition matrix with g goals $\mathbf{T}^{g+2 \times g+2}$ defined as:

$$\mathbf{T} = \begin{bmatrix} 1-\alpha & 0 & \dots & \alpha & 0 \\ 0 & 1-\alpha & \dots & \alpha & 0 \\ \vdots & & \ddots & & \vdots \\ \beta & \beta & \dots & 1-g\beta-\gamma & \gamma \\ 0 & 0 & \dots & \delta & 1-\delta \end{bmatrix}, \quad (3)$$

This transition matrix corresponds to the hidden Markov model (HMM) architecture visible in Fig. 3. The parameter $\alpha$ captures the worker tendency to change their mind, while the parameter couple $\beta$ and $\gamma$ set the threshold for estimating intention for each goal location. Increasing $\beta$ leads to quicker inference of worker's intentions and increasing $\gamma$ speeds up the decision making process. Parameter $\delta$ captures model's reluctance to return to estimating the other goal probabilities once it estimated that the worker is irrational. We performed several tests to determine the optimal values of these parameters which will be described in the "Experiments" section.

The worker intention is estimated using the Viterbi algorithm [5], which takes as inputs the hidden states set $S = \{G_1, ... G_g, G_?, G_x\}$, hidden state transition matrix $\mathbf{T}$, initial state $\Pi$, sequence of observations $\mathbf{O}$, and the emission matrix $\mathbf{B}$.

The emission matrix $\mathbf{B}$ is calculated using the motion validation vector $\mathbf{v}$. Since the observation is the validation vector $\mathbf{v}$ with continuous element values, the input to the Viterbi algorithm was modified by introducing an expandable emission matrix $\mathbf{B}^{k \times g}$, where $k$ is the recorded number of observations, are functions of the observation value. Once a new validation vector $v$ is calculated, the emission matrix is expanded with the row $\mathbf{B}'$, where the element $\mathbf{B}'_i$ stores the probability of observing $\mathbf{v}$ from hidden state $G_i$. The average of the last $m$ vectors $\mathbf{v}$ is also calculated and the maximum average value $\phi$ is selected. It is used as an indicator if the worker is behaving irrationally, i.e., is not moving towards any predefined goal. The value of the hyperparameter $m$ indicates how much evidence is to be collect before the worker is declared irrational. If the worker has been moving towards at least one goal in the last $m$ iterations ($\phi > 0.5$), $\mathbf{B}'$ is calculated as:

$$B' = \zeta \cdot \begin{bmatrix} \tanh(\mathbf{v}) & \tanh(1-\Delta) & 0 \end{bmatrix}, \quad (4)$$

and otherwise as:

$$B' = \zeta \cdot \begin{bmatrix} \mathbf{0}_{1 \times g} & \tanh(0.1) & \tanh(1-\phi) \end{bmatrix}, \quad (5)$$

where $\zeta$ is a normalising constant and $\Delta$ is calculated as the difference of the largest and second largest element of $\mathbf{v}$.

Finally, the initial probabilities of worker's intentions are set as:

$$\Pi = \begin{bmatrix} 0 & \dots & 0 & 1 & 0 \end{bmatrix}, \quad (6)$$

indicating that the initial state is $G_?$ and the model does not know which goal the worker desires the most. The Viterbi algorithm outputs the most probable hidden state sequence and the probabilities

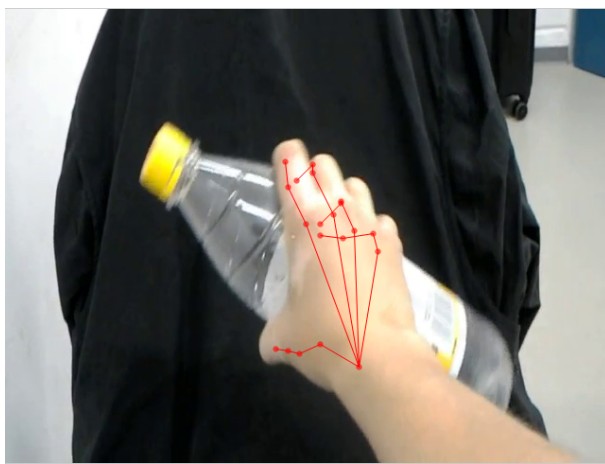

**Figure 4: The result of hand detection presented in [7] on HoloLens RGB camera data. One can see robust performance even during object handling.**

$P(G_i)$ of each hidden state in each step. These probabilities are the worker's intention estimates.

Goals can be added and removed during runtime as well making such a intention estimation framework quite flexible.

## 2.4 Action Estimation

Though estimating the goal of the human motion is extremely important for replanning robot motions to keep the interaction both safe and efficient, estimating actions the human wishes to perform could also bring additional information and flexibility to intention estimation systems.

Although the first generation of the HoloLens possesses inbuilt hand-tracking capabilities, these are quite limited and only four gestures can be tracked and classified. For a more robust hand following and classification we expanded the hand tracking by using the work presented in [7] on the HoloLens' RGB camera data. The algorithm tracks 21 hand joints and works with occlusions, surface contacts and object handling.

The detected hand joints are to classify actions - intention to grasp an object, grasped an object, pointing and stop. More actions can be classified in the future. The stop and pointing gestures are used as simple cues to control the robot. In addition, common gestures of fear or distress shall be classified as stop gestures, allowing the system indirect reaction to stress.

We presently only detect and use the right hand, however with a slight overhead the algorithm can detect both provided there is no significant overlap between them.

## 2.5 Robot Intention visualisation

The benefits of HMDs extend also to visualising robot intention. Instead of adapting robot motions to make them more legible [4], one can use holograms to signal the desired goal. In [15] it was shown that holographic information is adequate to show the goal of the robot, and even solve ambiguities if intention is expressed via synthesised voice. General motion intent can also be effectively

visualised using holographic cues [14]. In our work we chose to indicate the goal via a hologram containing a 3D sound source (spatial sound), as well as virtual execution - having a hologram of the robot execute the motion before the real robot performs it, such as shown in Fig. 5.

## 3 EXPERIMENTS

The experiments were aimed at testing the performance of the goal intention estimation. We used three goals in a circular pattern, from left to right - a green cylinder, a red cube and a blue sphere, as shown in Fig. 7.

The first set of experiments was aimed to find the optimal set of parameters $\alpha$, $\beta$, $\gamma$ and $\delta$ for our use case. Here we looked at the goal states and the transitions between them. The path was a simple left to right one, first going towards the green cylinder, then the red cube and finally the blue sphere. Figure Fig. 6 shows the behaviour of the parameter $\alpha$. A low value makes the algorithm too certain and almost does not spend time in the unknown state, while a high value makes the estimated goals jump too much. The optimal value of the parameter $\alpha$ was therefore set to $\alpha = 0.3$.

With the parameter $\alpha$ set, we tested the behaviour of changing the parameter $\beta$. A low $\beta$ maintains the unknown intention state too long, while a high $\beta$ lowers the general certainty but eliminates the insecurities between state transitions which is an unwanted behaviour. The value of $\beta$ was set to $\beta = 0.05$. In Fig. 8 one can see the behaviour of changing the parameter $\beta$.

The parameters $\gamma$ and $\delta$ did not significantly influence the outputs and were kept at the same value as in [10], namely $\gamma = 0.05$ and $\delta = 0.1$.

With the parameters set we examined how the algorithm behaves with different sequences of goals. The results are visible in Fig. 9.

The first test on the left is is a simple sequence of goals from left to right. One can notice the algorithm goes into the state of unknown intention during the transitions. The slower the transition the longer the unknown state. One can also see the small drop near the end when the hand tracking was lost and the gaze was not directly towards the sphere.

The second experiment starts with the middle goal, the cube, then goes left to the cylinder, back to the cube and then towards

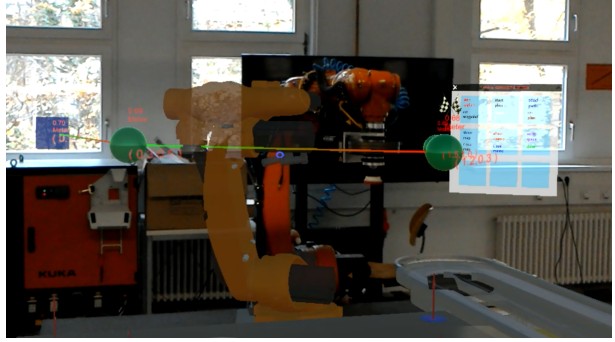

**Figure 5: An example of visualisation of the intentions of the robot to the human coworker - virtual execution with a holographic robot and plan visualisation.**

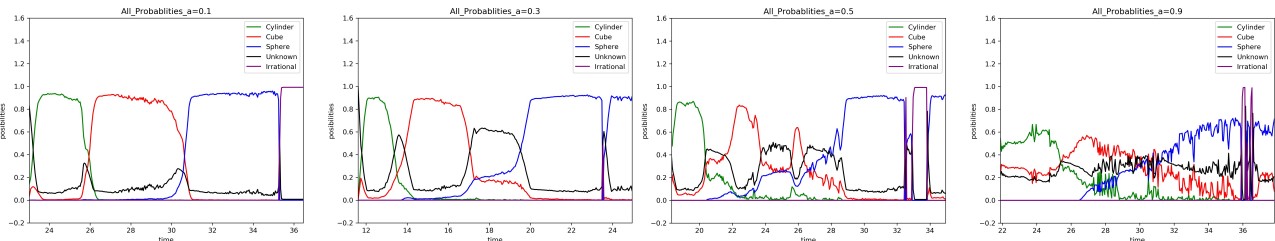

**Figure 6: The effect of increasing the parameter $\alpha$. The parameter captures the worker tendency to change their mind. A low $\alpha$ will make the algorithm "too sure" about the intention, while a too high $\alpha$ produces a chaotic and unusable output.**

the sphere. One can see that the transition from cylinder to cube lasts slightly longer than from cube to cylinder. This is due to the fact that the algorithm is reluctant to estimate an already visited or skipped goal. One can also see the long transition between the cube and the sphere, as the algorithm prefers the goal that has already been visited two times. This shows that the estimation follows our intuition.

Finally, the third experiment shows what happens when the user does a complete rotation and faces away from all of the three goals. Again the algorithm performs quite intuitively and proclaims the user "irrational" as all the possible goals were completely on the other side.

Additionally, we tested simple interactions between an industrial manipulator and the human user. In the first one the robot was selecting goals and randomly. Should the goal intention estimation detect that the human is moving to the same goal the robot would stop and select a new goal. Additionally we used the same framework to navigate the manipulator to the estimated goal of the human, proving that the framework can also be beneficial in teleportation scenarios.

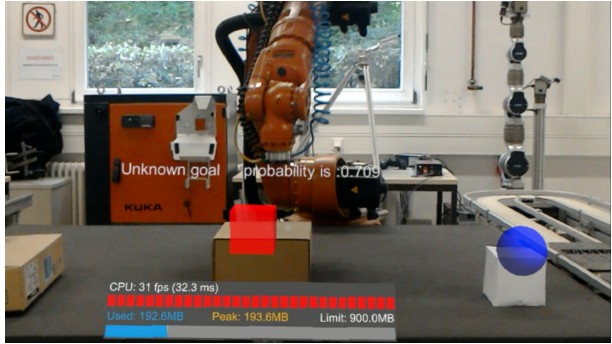

**Figure 7: The interaction setup with an industrial robot, the three spatial goals are represented by colour and shape. In this experiment the user took direct control of the robot and the human intention estimation is used to detect to which object does the user wish the robot to move, illustrating another use case of intention estimation.**

## 4 CONCLUSION AND IMPACT

In this work we presented a completely portable, robot agnostic system for intention estimation and visualisation for human-robot collaboration scenarios. Our system does not require any special set-up or sensors on or around the robot and is capable of both estimating the human coworkers goals and actions as well as visualise the goals and intentions of the robot coworker.

Having an intention estimation system, in addition to explicit intention declarations, can greatly reduce the mental and physical workload on the user, while providing constant, information rich data to the robot, thereby improving the safety and efficiency of robot motions.

We have shown that the goal prediction part of the HAIR system works as intended and indeed the goal intentions estimated follow a reasoning that humans might find intuitive and agree with.

Predicting the goal and motion of the human coworker can increase both safety for the human and the efficiency of robot motions. The goal estimation system [10] was integrated into a mobile robot fleet management system of a simulated automated warehouse. In [9] it was shown that the proposed system markedly improved warehouse efficiency compared to no goal estimation or even a simplistic one. It is to be expected that such an efficiency increase would also be observed in interactions with a robotic manipulator. Further testing is going to be needed however, to support that claim.

Likewise the action estimation component as well as the entire system needs to be evaluated in user studies. More specifically the change in mental and physical workloads between various intention sharing modalities is of great interest and quite important in proving the claims that the intention estimation algorithms presented here significantly decrease the workload compared to explicitly stating the goals.

As HMDs become ever more common, and the amount of robot coworkers per human coworker continues to increase, having intuitive HRI using systems that are cheap, simple and portable becomes essential. Lowering the complexity and price of each robot by exploiting wearables will lead to a wider use of robots and increased human-robot collaboration. We hope that the research presented here provides the first stepping stones towards such a system. HAIR should raise the quality of interaction between robots and humans, instead of such interactions raising the hair on the necks of the human coworkers.

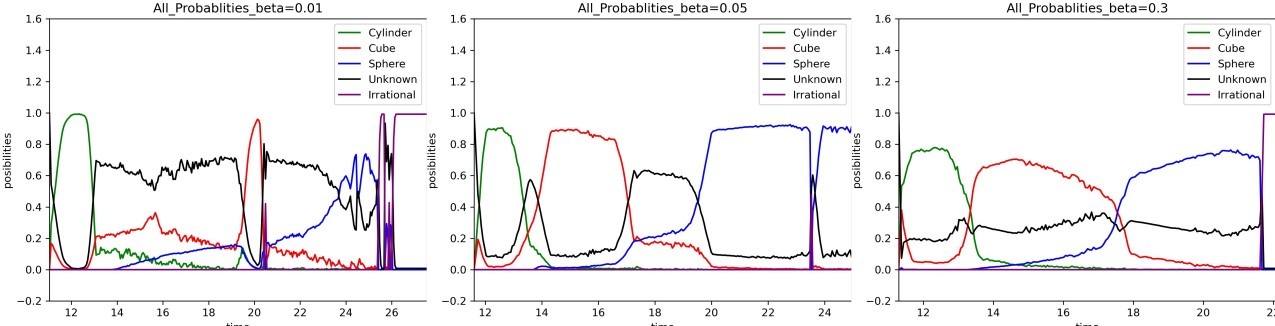

**Figure 8: The effect of increasing the parameter $\beta$. The parameter couple $\beta$ and $\gamma$ set the threshold for estimating intentions for each goal location. A low $\beta$ will make the algorithm estimate the unknown intention too much rendering it unusable , while a too high $\beta$ lowers the general certainty but eliminates the insecurities between transitions which is an unwanted behaviour.**

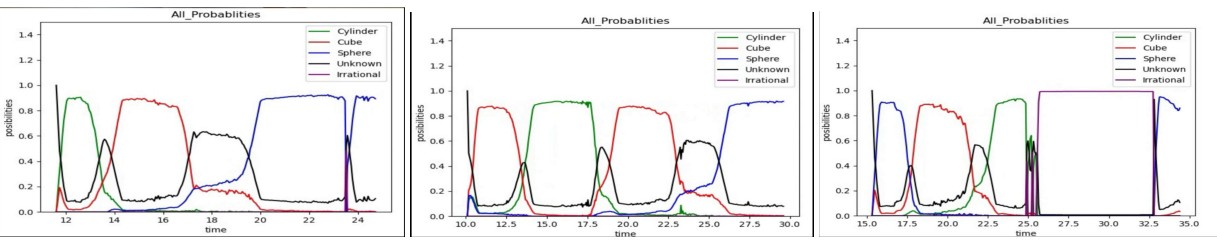

**Figure 9: Three tests with different goal order. On the left, the user was selecting goals left to right - cylinder, cube then sphere. In the middle the user starts with the cube, moves to the cylinder, back to the cube and finally goes to the sphere. In the test on the right the starting goal is sphere, then cube, then cylinder, after which the user turns around completely and ends back on the sphere.**

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
