# OpenReview forum: "HAIR: Head-mounted AR Intention Recognition"
_humanrobotinteraction.org/HRI/2021I/Workshop/VAM-HRI — VAM-HRI 2021 Oral_

### Official Review · AnonReviewer2 · 2021-03-04
**This paper is easy to read, approaches an interesting problem, and provides a good technical foundation**

**Rating:** 8
**Confidence:** 5

**Review:**

This paper presents a method of integrating intention recognition and portable head-mounted displays, along with visualizations for communicating robot intention. The method automatically infers the human’s intention (represented as a goal pose in space) based on multimodal information like eye-gaze and hand gestures. In order to make reasonable inferences based on the input, this paper proposes an observation model for gaze and actions and uses a hidden markov model framework to probabilistically fuse observations together to make estimates of the latent goal state. This information is then visualized to the user to communicate the robot's mental state. In an initial set of experiments, the authors aimed to find a good setting of hyper parameters for their model to ensure inference was quick but still provided the ability to remove from noise. This method is interesting because it provides a portable system for estimating and visualizing human intention with an augmented reality headset.

Overall, this paper is easy to read, approaches an interesting problem, and provides a good technical foundation (HMMs) to perform probabilistic inference with. The figures are helpful and the math is easy to follow. I recommend this paper be accepted.

Comments:
- This paper is missing an important work that was previously done on inferring and visualizing human intent for human-robot collaboration using AR headsets (Eric Rosen, David Whitney, Michael Fishman, Daniel Ullman, Stefanie Tellex. "Mixed Reality as a Bidirectional Communication Interface for Human-Robot Interaction". 2020 IEEE/RSJ International Conference on Intelligent Robots and Systems (IROS). IEEE, 2020.). This work is extremely related and should be mentioned, since they use a generalization of the HMM framework to represent human intent and infer it through multimodal information like gaze, language, and hand gestures, and visualize this information to the user (they represent the problem as a partially observable markov decision process to jointly enable intent inference and robot decision making)
- What do the authors think about the fact eye-gaze plays two roles: to convey intention, but also to learn about the environment. Would it be possible to  differentiate between these two modes so that certain observations of eye-gazes are not included in updating inferred human intent?

---

### Official Review · AnonReviewer3 · 2021-03-04
**HAIR Review**

**Rating:** 8
**Confidence:** 5

**Review:**

This paper discusses an intention estimation AR program for human intention (goals) within a human-robot team task (manipulation arm). The system implements object detection from the HMD while using an HMM leveraging hand tracking to distinguish intent. The paper then described experiments completed along with differing alpha and beta values. The reported results indicate the ability to distinguish human goal intent along with robot visualization via the HMD.

This paper is very interesting and could make for interesting use within human-robot teaming.For Experiments, who is/are the participants? I think it is fine if all the data are from one person for initial work but it should be stated somewhere. I would also love to see a video and/or open-source code if at all possible.


Other comments:

For all figures from Hololens perspective, you may want to look at removing the performance visualizer (https://microsoft.github.io/MixedRealityToolkit-Unity/Documentation/Performance/PerfGettingStarted.html)

In sentence 2, change “development” to “develop”

---

### Decision · Program_Chairs · 2021-03-06

Accept (Oral)